# Prognostic Significance of Volumetric Parameters Based on FDG PET/CT in Patients with Lung Adenocarcinoma Undergoing Curative Surgery

**DOI:** 10.3390/cancers15174380

**Published:** 2023-09-01

**Authors:** Hyunjong Lee, Yoon-La Choi, Hong Kwan Kim, Yong Soo Choi, Hojoong Kim, Myung-Ju Ahn, Hong Ryul Pyo, Joon Young Choi

**Affiliations:** 1Department of Nuclear Medicine, Samsung Medical Center, Sungkyunkwan University School of Medicine, Seoul 06351, Republic of Korea; nmhjlee@gmail.com; 2Department of Pathology, Samsung Medical Center, Sungkyunkwan University School of Medicine, Seoul 06351, Republic of Korea; yla.choi@samsung.com; 3Department of Thoracic and Cardiovascular Surgery, Samsung Medical Center, Sungkyunkwan University School of Medicine, Seoul 06351, Republic of Korea; hkts.kim@samsung.com (H.K.K.); ysooyah.choi@samsung.com (Y.S.C.); 4Division of Pulmonary and Critical Care Medicine, Department of Medicine, Samsung Medical Center, Sungkyunkwan University School of Medicine, Seoul 06351, Republic of Korea; hj3425.kim@samsung.com; 5Division of Hematology-Oncology, Department of Medicine, Samsung Medical Center, Sungkyunkwan University School of Medicine, Seoul 06351, Republic of Korea; silk.ahn@samsung.com; 6Department of Radiation Oncology, Samsung Medical Center, Sungkyunkwan University School of Medicine, Seoul 06351, Republic of Korea; hr.pyo@samsung.com

**Keywords:** lung cancer, adenocarcinoma, FDG PET/CT, prognosis, metabolic tumor volume, total lesion glycolysis

## Abstract

**Simple Summary:**

Lung cancer is a leading cause of death worldwide, and it’s crucial to know severity of cancer and predict its prognosis. Our study focused on using a special type of scan known as FDG PET/CT to predict how well patients with a specific type of lung cancer, called adenocarcinoma, would fare after surgery. The study looked at 432 patients and analyzed various factors like age, sex, and certain imaging parameters of the tumors. Our key finding was that the metabolic tumor volume of tumor was powerful indicator of prognosis after surgery. This information was even more accurate when combined with TNM staging, existing methods used to stage cancer. Therefore, these scans could provide doctors with valuable information to plan better treatments. This development has the potential to improve survival rates and quality of life for lung adenocarcinoma patients.

**Abstract:**

Introduction: FDG PET/CT is a robust imaging modality to diagnose and stratify prognoses for non-small cell lung carcinoma. However, the role of FDG PET/CT in operable lung adenocarcinoma patients has not been previously investigated in a large cohort with varying pathological stages. The prognostic value of volumetric parameters based on FDG PET/CT was investigated in patients with stage I–III lung adenocarcinoma receiving curative surgery. Methods: This retrospective study included 432 patients with lung adenocarcinoma undergoing preoperative FDG PET/CT between January 2016 and December 2017. Clinicopathologic variables, conventional image parameters, such as the maximum standardized uptake value (SUVmax) and mean SUV (SUVmean) of the primary tumor, and volumetric parameters, such as metabolic tumor volume (MTV) and total lesion glycolysis (TLG), were included in Cox regression analysis. Subgroup analysis was conducted to compare hazard ratios (HRs) based on MTV in each pathological stage. A new staging system including volumetric parameters was suggested. Results: A total of 432 patients (median age: 62 years; interquartile range: 56–70 years; 225 males) were evaluated. Sex, age, presence of EGFR mutation, pathological stage, MTV, and TLG of the primary tumor were selected as statistically significant prognostic factors for overall survival irrespective of other variables (OS; *p* < 0.05 for all). Pathological stage, MTV, and TLG of the primary tumor were selected as statistically significant prognostic factors for disease-free survival irrespective of other variables (*p* < 0.05 for all). The suggested new staging system including MTV as an additional criterion showed better discrimination and prediction for OS than the conventional pathological staging system despite statistical insignificance (concordance index: 0.698 vs. 0.673). Conclusions: The volumetric parameters of the primary tumor based on preoperative FDG PET/CT were independent prognostic factors in addition to pathological stage in patients with operable lung adenocarcinoma. The suggested new staging system considering MTV predicted the prognoses better than the conventional pathological staging system.

## 1. Introduction

In 2020, the global incidence of lung cancer was the second highest of all cancers, with the highest mortality [1]. Lung cancer is classified into two pathologic groups: small cell lung carcinoma and non-small cell lung carcinoma (NSCLC). In the National Comprehensive Cancer Network (NCCN) guideline, surgical resection is recommended first for early-stage or clinically N0 lung cancer. For lung cancer with positive margins or that has spread to regional lymph nodes, adjuvant radiotherapy or chemotherapy after resection is generally recommended [2]. In a previous study, recurrent tumors developed in 38% of stage I lung cancer patients after resection [3]. Therefore, determining prognostic markers to predict which patients with operable lung cancer are at risk of poor prognosis is important, and closer surveillance or adjuvant therapy may be required for those patients.

F-18 fluorodeoxyglucose positron emission tomography/computed tomography (FDG PET/CT) is a useful imaging modality to evaluate the stage of and select the appropriate treatment option for lung cancer [4]. Maximum standardized uptake value (SUVmax), the most representative parameter based on FDG PET/CT, provides a threshold value to diagnose malignant nodules in the lung and is a good prognostic factor for overall survival (OS) in lung cancer [5,6]. In addition to SUVmax, volumetric parameters, such as metabolic tumor volume (MTV) and total lesion glycolysis (TLG), have exhibited good prognostic power in lung adenocarcinoma [7,8,9]. Furthermore, dual-time-point studies have recently been highlighted for exploring the biology of lung cancer and diagnostics related to lung nodules [10,11]. Although many studies have been conducted, they included a small number of patients or used meta-analysis methods.

In particular, studies are needed in which the focus is on prognosis prediction in adenocarcinoma. First, adenocarcinoma is the most common histologic type in NSCLC [12]. In addition, metabolic pattern and characteristics represented on FDG uptake differ between adenocarcinoma and squamous cell carcinoma [13,14,15]. Differences in prognosis and treatment options, such as targeted therapy, increase the necessity of separate analysis for adenocarcinoma [16,17]. However, analysis of only adenocarcinoma is seldom performed in most studies. More evidence is still required to determine the prognostic value of volumetric parameters compared with conventional image parameters or other clinical factors in lung adenocarcinoma patients.

In the present study, the prognostic value of FDG PET/CT was investigated in a large number of subjects with operable lung adenocarcinoma. The significance of MTV as a reference to stratify patients with the same pathological stage was also determined. Finally, a new staging classification reflecting a volumetric parameter was proposed and its prognostic power evaluated.

## 2. Methods

### 2.1. Subjects

A total of 874 consecutive patients undergoing FDG PET/CT examination for the initial staging of lung adenocarcinoma and subsequent curative surgery between January 2016 and December 2017 were retrospectively included in the present study. The following patients were excluded: 385 subjects with a primary tumor with SUVmax less than 2.5, the threshold to calculate volumetric parameters, 18 patients with pathologic T4 stage due to the small number of subjects, 17 patients without pathologic information of histologic grade, 17 patients without information on EGFR mutation, and 5 patients without information on ALK mutation. Finally, 432 patients were included in the present study (Figure 1). All the patients underwent radiologic examination, such as chest X-ray and contrast-enhanced chest CT, for preoperative initial staging workup, as well as FDG PET/CT. The institutional review board of our institution approved this retrospective cohort study (IRB #2022-01-086) and informed consent was waived.

The clinical characteristics and demographics of the subjects are described in Table 1. Overall, 225 patients were male (52.1%). The median age was 62 years (interquartile range: 56–70 years). The median SUVmax was 6.4, SUVmean 3.1, MTV 6.3 cm^3^, and TLG 19.8. The tumors were located in the left lung in 44.2% of subjects. Adjuvant therapy was implemented for 41.2% of patients after curative surgery. Pathological findings showed 64.6% of patients had no lymph node metastasis. Stage IA was the most common stage among pathological substages (28.5%). Stage I was the most common among pathological stages (45.1%).

### 2.2. FDG PET/CT Acquisition and Analysis

All patients fasted for at least 6 h and had blood glucose levels less than 200 mg/dL at the time of their FDG PET/CT scans. Whole-body PET and CT images from the base of the skull to mid-thigh were acquired 60 min after the injection of 5.0 MBq/kg FDG without intravenous or oral contrast on a Discovery LS or a Discovery STE PET/CT scanner (GE Healthcare, Milwaukee, WI, USA). Continuous spiral CT was performed with an 8-slice helical CT (140 keV, 40–120 mA; Discovery LS) or 16-slice helical CT (140 keV, 30–170 mA; Discovery STE). An emission scan was then obtained from head to thigh for 4 min per frame in two-dimensional mode (Discovery LS) or 2.5 min per frame in three-dimensional mode (Discovery STE). PET images were reconstructed using CT for the attenuation correction using the ordered-subsets expectation maximization algorithm with 28 subsets and 2 iterations (matrix 128 × 128, voxel size 4.3 × 4.3 × 3.9 mm; Discovery LS) or ordered-subsets expectation maximization algorithm with 20 subsets and 2 iterations (matrix 128 × 128, voxel size 3.9 × 3.9 × 3.3 mm; Discovery STE). The image acquisition protocol was designed in accordance with international guidelines, and technical conditions are summarized in Appendix A. PET/CT images of 137 patients (31.7%) were acquired using Discovery LS and 295 patients with Discovery STE.

Image parameters were acquired on the threshold segmentation method with a threshold SUV value of 2.5 in MIM version 6.4 software (MIM Software Inc., Cleveland, OH, USA). Briefly, the target primary tumor was identified by an experienced nuclear medicine physician who was blinded to all clinical information except the target tumor site. As the physician dragged the cursor out from the center of the target tumor to a point near the edge of the lesion, the software automatically outlined a three-dimensional volume of interest with SUV greater than 2.5 on the tumor. SUVmax, SUVmean, MTV, and TLG were calculated from the segmented target tumor lesion.

### 2.3. Clinicopathologic Variables and Clinical Follow-Up

Clinical information including sex, age, adjuvant therapy, histological type, presence of EGFR or ALK mutation, and pathologic reports was obtained by reviewing electronic medical records. The pathological TNM stage and substage were determined based on the eighth edition of the American Joint Committee on Cancer (AJCC)/Union for International Cancer Control (UICC) staging system. Adjuvant therapy was implemented after surgery based on each patient’s situation and their corresponding physician’s decision. After surgery, all patients were regularly monitored to obtain accurate information regarding recurrence or metastasis. The follow-up program was every 3–6 months during the first to second year and every year thereafter. Every follow-up evaluation included a complete physical examination, complete blood count, biochemical screening, and chest X-ray. CT scans of the chest were performed from every 6 months to 1 year, or more frequently if clinically indicated.

Recurrence or metastasis was considered when an abnormal finding indicating recurrence or metastasis was found on serial imaging studies or pathologically confirmed malignancy. The events for survival analysis were defined as recurrence or metastasis and any cause of death. The disease-free survival (DFS) and OS durations from the last follow-up or event were recorded for each patient.

### 2.4. Statistical Analysis

Age as a continuous scale was divided into three groups as a discrete scale based on tertiles for log-rank tests and multivariate analyses. Image parameters including SUVmax, SUVmean, MTV, and TLG as a continuous scale were divided into two groups as a discrete scale based on a cutoff value to best discriminate the prognosis of OS in all patients. The cutoff values were explored using the “surv_cutpoint” function in the “survminer” package. The “surv_cutpoint” function determines the optimal cutoff for continuous variables, using the maximally selected rank statistics from the “maxstat” package. Cutoff values were applied for log-rank tests and multivariate analyses as well as for suggestion of a reference value for a new staging system. Clinical variables including sex, age with both discrete and continuous scales, location of primary tumor, performance of adjuvant therapy, histological grade of primary tumor, pathological stage, presence of EGFR or ALK mutation, SUVmax, SUVmean, MTV, and TLG were used for univariate survival analysis. Both OS and DFS were endpoints of analysis. The Cox proportional hazards model was used to evaluate the prognostic power of each variable. HRs and 95% confidence intervals (CIs) were estimated. Log-rank statistics were also obtained using the Kaplan–Meier method. Significant variables in univariate survival analysis with P-values from log-rank statistics less than 0.05 were included in multivariate survival analysis. Due to multicollinearity, multivariate survival analysis was repeatedly performed for each image parameter.

Patients in each pathological stage (I, II, and III) were divided into two groups based on the reference values of MTV. HRs and 95% CIs were estimated and log-rank statistics were obtained using the Kaplan–Meier method. Subgroups showing no significant difference in HRs were classified into the same group. Kaplan–Meier analysis was conducted in the newly classified stages, which were also included in multivariate analysis with other significant variables in univariate survival analysis instead of conventional pathological stage. To compare prognostic power of the new staging system with the conventional staging system, time-dependent receiver operating characteristic (ROC) curves were plotted, an efficient tool for measuring or comparing the performance of survival models given the survival or disease status of individuals at certain time points. Time-dependent ROC curves were represented using the “timeROC” function in the package “timeROC”. Subsequently, areas under the curve (AUCs) of each ROC curve were calculated using the same function. All statistical analyses were performed using R software (v. 4.0.4, R Foundation for Statistical Computing, Vienna, Austria). A *p*-value less than 0.05 was considered statistically significant.

## 3. Results

### 3.1. Survival Analysis Data

In univariate survival analysis, sex, age with discrete scale, age with continuous scale, adjuvant therapy, histological grade, EGFR mutation, pathological T stage, pathological N stage, pathological substage, pathological stage, SUVmax with discrete scale, SUVmax with continuous scale, SUVmean with discrete scale, SUVmean with continuous scale, MTV with discrete scale, MTV with continuous scale, TLG with discrete scale, and TLG with continuous scale were significant prognostic factors for OS (*p* < 0.05 for all, Table 2). Adjuvant therapy, pathological T stage, pathological N stage, pathological substage, pathological stage, SUVmax with discrete scale, SUVmax with continuous scale, SUVmean with discrete scale, SUVmean with continuous scale, MTV with discrete scale, MTV with continuous scale, TLG with discrete scale, and TLG with continuous scale were significant prognostic factors for DFS (*p* < 0.05 for all, Appendix A).

Due to multicollinearity issues, multivariate survival analysis was repeatedly performed based on each image parameter. In the multivariate survival analysis, age, EGFR mutation, and pathological stage were independent prognostic factors for OS among clinicopathologic variables (*p* < 0.05 for all). Among image parameters, MTV and TLG were selected as independent prognostic factors for OS (HR 2.043 and 1.761, respectively; Table 3). On the contrary, SUVmax and SUVmean showed no statistical significance (*p* = 0.106 and *p* = 0.124, respectively; Appendix A). For DFS, SUVmax, SUVmean, MTV, TLG, and pathological stage were selected as independent prognostic factors in the multivariate survival analysis (Appendix A). Survival curves of pathological stage, MTV, and TLG are shown in Figure 2 and Figure 3.

### 3.2. Proposed New Staging System including MTV

Because MTV was the most significant prognostic factor for OS in multivariate analysis, subgroup analysis was performed to investigate the role of MTV in each stage and adjust the staging system to better discriminate prognoses. HRs for OS were calculated in each subgroup classified based on each pathological stage and MTV group. The HRs of stage I subjects with low MTV and stage II subjects with low MTV did not significantly differ (*p* = 0.165, Table 4). Similarly, the HRs of stage I subjects with high MTV and stage II subjects with high MTV, and stage III subjects with low MTV, did not significantly differ (*p* > 0.05 for all). The new stages were classified based on these results. Stage II subjects with low MTV were downstaged into a new stage I. Stage I subjects with high MTV were upstaged into a new stage II. Stage III subjects with low MTV were downstaged into a new stage II.

Although the HRs for stage I and stage II subjects showed significant difference in the conventional staging system, HR of stage II was higher in the proposed new staging system (HR = 2.375 and *p* = 0.006 vs. HR = 3.128 and *p* < 0.001, respectively; Appendix A). Compared with survival curves based on the conventional staging system, curves based on the improved staging system discriminated prognoses better (concordance index = 0.673 and 0.698, respectively; Figure 4). Time-dependent ROC curves for OS were calculated to compare the superiority of the new staging system with the conventional staging system (Figure 5). Time-dependent ROC curves based on the Cox regression model showed the new staging system had higher AUC values than the conventional staging system (Figure 5A,C). Other independent prognostic factors, such as age and EGFR mutation, were additionally included in the Cox regression model for time-dependent ROC curves of the new and conventional staging system. Similarly, time-dependent ROC curves based on the Cox regression model showed the new staging system had higher AUC values than the conventional staging system despite statistical insignificance (Figure 5B,D). Figure 6 shows two representative cases upstaged or downstaged based on the MTV of the primary tumor.

## 4. Discussion

The present study showed the MTV and TLG of the primary tumor on preoperative FDG PET/CT in operable lung adenocarcinoma patients were independent prognostic factors in multivariate analysis but not SUVmax and SUVmean. When subjects were grouped into the new staging system that included the pathological stage and MTV, discriminating and predicting prognoses was improved compared with the conventional pathological staging system.

In previous studies, the prognostic value of FDG PET/CT in NSCLC was reported. SUV was a good prognostic factor for OS in NSCLC [6,18]. SUVmax is the most representative parameter in FDG PET/CT. Despite the adequate usability and significance in the clinical field, SUVmax cannot represent the total metabolic feature of the entire tumor mass because it is a value from a single voxel in nonhomogeneous tumor tissue. Volumetric PET parameters, such as MTV and TLG, are widely used in clinical studies due to their superiority to reflect tumor burden. In the present study, MTV and TLG but not SUVmax showed prognostic significance in NSCLC, which is consistent with previous studies [8,19].

The present study had several advantages compared with previous studies. First, inclusion of only adenocarcinoma was rare in most of the previous studies, despite significant difference in FDG uptake compared with squamous cell carcinoma [14]. Due to differences in metabolism and oncogenic mechanism between adenocarcinoma and squamous cell carcinoma, the FDG uptake can significantly differ between the two histological types. Second, a small number of patients were enrolled in previous studies in which only lung adenocarcinoma patients were evaluated; Chung et al. analyzed 106 patients [7], Ouyang et al. 157 patients [20], and Kim et al. 102 patients [21]. To the best of our knowledge, the present study is the first in which the prognostic value of volumetric parameters on FDG PET/CT was determined in a large cohort of only lung adenocarcinoma subjects.

In the present study, only lung adenocarcinoma tumors with SUVmax greater than 2.5 were selected for analysis. A threshold of 2.5 for tumor segmentation was chosen for two reasons. First, 2.5 is a widely accepted threshold value to discriminate malignancy and predict prognosis in NSCLC [22,23]. Second, other segmentation methods, such as the gradient-based method, have several disadvantages [24] and may be sensitive to the acquisition protocols or reconstruction parameters of the PET images. In addition, other methods cannot provide segmentation of inner hollow structures such as central necrosis. Because the present study included PET images from two different instruments, tumor segmentation with SUV 2.5 was determined more appropriate than other methods.

In the clinical field, TNM stage is widely used to select appropriate treatment options for various malignancies including NSCLC. The NCCN guideline recommends adjuvant chemotherapy for patients with high-risk stage IB or stage IIA NSCLC [2]. The guideline suggests the differentiation grade, presence of vascular invasion, tumor size, and pleural involvement as criteria for risk classification. Based on the results of this study, the MTV of the primary tumor is another useful criterion to select high-risk stage I patients due to its satisfactory prognostic predictability and the non-invasiveness of FDG PET/CT examination. The follow-up intervals for subjects with high MTV should be shorter than the intervals for individuals with low MTV considering the high risk of poor DFS and OS. In brief, patients with the same TNM stage can be treated with additional treatment options and follow-up plans based on the MTV of the primary tumor.

This present study had several limitations. First, two different PET/CT scanners were included in the study, which might affect the SUV measurement. However, in a previous study, SUVs from varying instruments did not differ in a phantom study [25]. Therefore, the analysis was performed without a specific integration process. Second, only subjects undergoing curative surgery and having accurate pathological stage information were included in the present study. Thus, a small number of subjects with high stages were included. Further studies including lung adenocarcinoma patients treated with other options, such as concurrent chemoradiotherapy, are warranted. Third, the design was limited to the new staging system based on pathologic substages due to a lack of subjects with specific substages such as stage IIA or IIIB. Further studies with more patients are warranted to support the value of volumetric parameters for upstaging or downstaging in pathological substages. Finally, selection bias may be introduced due to exclusion of a substantial number of patients due to low SUV. However, given that the prognostic value of FDG PET/CT may be diminished for these patients due to their inherently favorable prognosis, the overall significance of this study remains robust. Still, a further study employing a gradient-based segmentation method like ‘PETedge’ that incorporates even low-SUV lesions could provide more comprehensive insights.

## 5. Conclusions

The MTV and TLG of the primary tumor on FDG PET/CT were independent prognostic factors in addition to pathological stage in lung adenocarcinoma patients undergoing curative surgery. The proposed new staging system including MTV could better predict prognoses than the conventional TNM staging system. Closer follow-up or further adjuvant treatment should be considered for lung adenocarcinoma patients with high MTV even with the same pathological stage due to the high risk of poor OS.

## Figures and Tables

**Figure 1 cancers-15-04380-f001:**
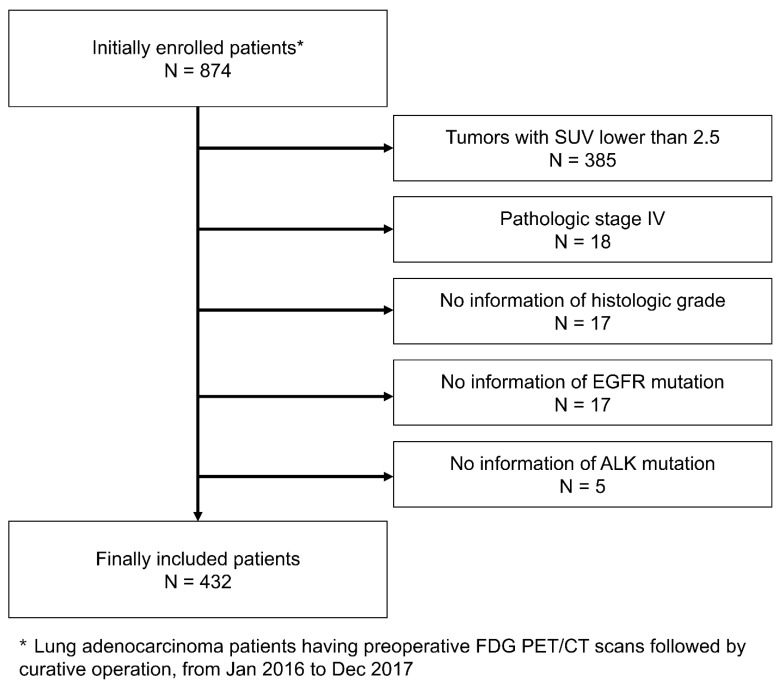
Patient inclusion and exclusion criteria. The present study included 874 retrospectively enrolled patients. Patients with a primary tumor with SUVmax less than 2.5 and pathologic T4 stage, without pathologic information on histologic grade, EGFR mutation, or ALK mutation, were excluded. Finally, 432 patients were included.

**Figure 2 cancers-15-04380-f002:**
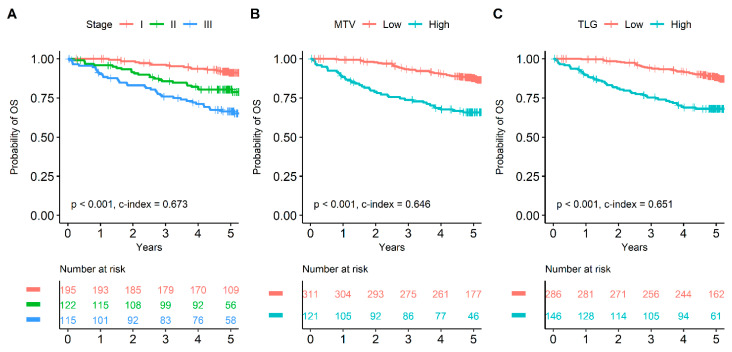
Survival curves for OS based on pathological stage, MTV, and TLG. Pathological stage adequately discriminated prognoses for OS (**A**). Patients with MTV greater than 12.7 cm^3^ showed significantly poorer prognosis than subjects with MTV less than 12.7 cm^3^ (**B**). Patients with TLG greater than 37.0 showed significantly poorer prognosis than subjects with TLG less than 37.0 (**C**).

**Figure 3 cancers-15-04380-f003:**
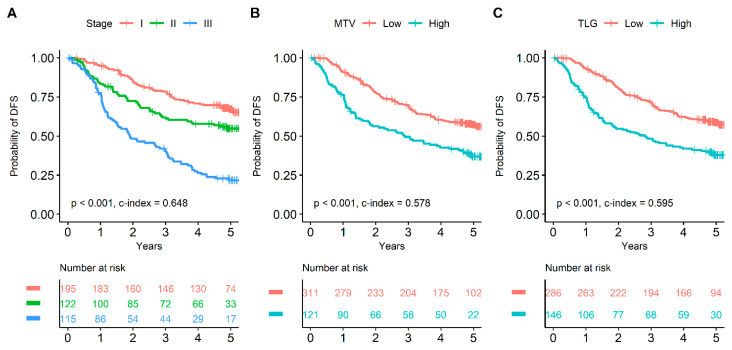
Survival curves for DFS based on pathological stage, MTV, and TLG. Pathological stage adequately discriminated prognoses for DFS (**A**). Patients with MTV greater than 12.7 cm^3^ showed significantly poorer prognosis than subjects with MTV less than 12.7 cm^3^ (**B**). Patients with TLG greater than 37.0 showed significantly poorer prognosis than subjects with TLG less than 37.0 (**C**).

**Figure 4 cancers-15-04380-f004:**
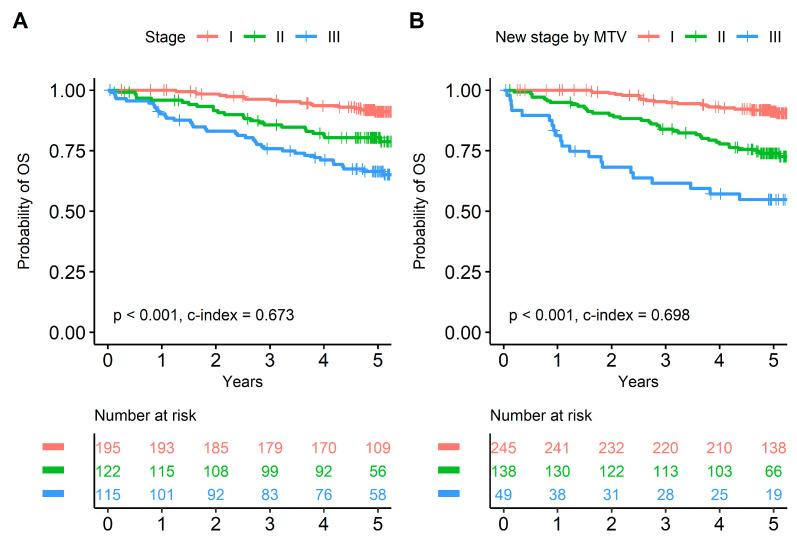
Survival curves based on the conventional stage and proposed new stage. Compared with survival curves based on the conventional pathological stage (**A**), curves based on the proposed new staging system including MTV discriminated prognoses better (**B**).

**Figure 5 cancers-15-04380-f005:**
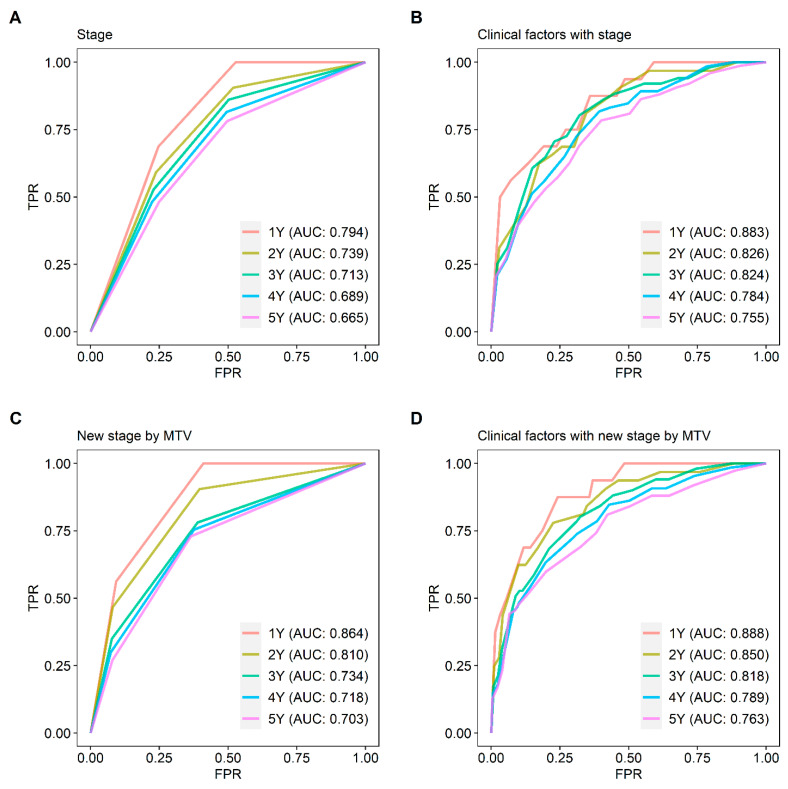
Time-dependent ROC curves based on the conventional stage and the proposed new stage. Time-dependent ROC curves with conventional staging system (**A**) showed lower AUC values than ROC curves with the proposed new staging system (**C**). Even when including other independent prognostic factors such as age and EGFR mutation, lower AUC values were observed with the conventional staging system (**B**) than AUC values with the proposed new staging system (**D**).

**Figure 6 cancers-15-04380-f006:**
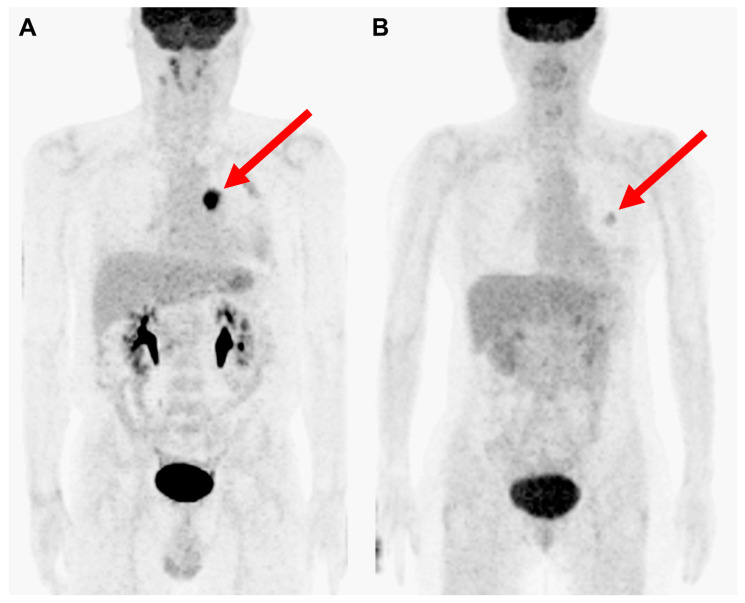
Representative cases of upstaging and downstaging based on MTV. Maximal intensity projection images of FDG PET/CT demonstrate representative cases of upstaging and downstaging. (**A**) A 70-year-old male patient with stage I lung adenocarcinoma. The tumor was clearly observed in the image with MTV of 30.7 cm^3^ (red arrow). Thus, the patient was upstaged into the new proposed stage II. DFS duration was 368 days and OS duration was 738 days. (**B**) A 56-year-old female patient with stage III lung adenocarcinoma. The tumor was poorly delineated in the image due to its low MTV of 0.3 (red arrow). Thus, the patient was downstaged into the new proposed stage II. OS duration was 1992 days without evidence of recurrence or death.

**Table 1 cancers-15-04380-t001:** Demographic and clinical characteristics of patients with operable lung adenocarcinoma.

Characteristics	Patients, *n* (%)
Sex	
Female vs. male	207 (47.9) vs. 225 (52.1)
Age, median (interquartile range), years	62 (56–70)
<58, 58–67, >67	143 (33.1), 143 (33.1), 146 (33.8)
SUVmax, median (range)	6.4 (2.5–32.6)
<4.4 vs. ≥4.4	137 (31.7) vs. 295 (68.2)
SUVmean, median (range)	3.1 (0.9–8.9)
<3.6 vs. ≥3.6	133 (30.8) vs. 299 (69.2)
MTV, median (range), cm^3^	6.3 (0.1–180.0)
<12.7 vs. ≥12.7	311 (72.0) vs. 121 (28.0)
TLG, median (range)	19.8 (0.2–1335.9)
<37.0 vs. ≥37.0	146 (33.8) vs. 286 (66.2)
Location	
Left vs. Right	191 (44.2) vs. 241 (55.8)
Adjuvant therapy	
No, C, CR, R	254 (58.8), 149 (32.4), 25 (5.8), 4 (0.9)
Histological grade	
WD, MD, PD	1 (0.2), 285 (66.0), 146 (33.8)
EGFR mutation	
No vs. Yes	233 (53.9) vs. 199 (46.1)
ALK mutation	
No vs. Yes	395 (91.4) vs. 37 (8.6)
Pathological T stage	
T_mi_, T1a, T1b, T1c, T2a, T2b, T3, T4	1 (0.2), 2 (0.5), 63 (14.6), 115 (26.6), 124 (28.7), 46 (10.6), 56 (13.0), 25 (5.8)
Pathological N stage	
N0, N1, N2	279 (64.6), 70 (16.2), 83 (19.2)
Pathological substage	
IA, IB, IIA, IIB, IIIA, IIIB	123 (28.5), 72 (16.7), 26 (6.0), 96 (22.2), 104 (24.1), 11 (2.5)
Pathological stage	
I, II, III	195 (45.1), 122 (28.2), 115 (26.6)

SUVmax, maximum standardized uptake value; SUVmean, mean standardized uptake value; MTV, metabolic tumor volume; TLG, total lesion glycolysis; C, chemotherapy; CR, chemoradiotherapy; R, radiotherapy; WD, well differentiated; MD, moderately differentiated; PD, poorly differentiated.

**Table 2 cancers-15-04380-t002:** Univariate Cox regression analysis for overall survival in patients with operable lung adenocarcinoma.

Variable	Categories	HR (95% CI)	*p*-Value	*p*-Value from Log-Rank Test
Sex	Female vs. male	1.783 (1.133–2.804)	0.012	0.011
Age, years	<58			<0.001
	58–67	1.715 (0.900–3.270)	0.101	
	>67	3.464 (1.923–6.238)	<0.001	
Age (continuous)		1.054 (1.030–1.078)	<0.001	-
Location	Left vs. Right	0.666 (0.423–1.048)	0.079	0.077
Adjuvant therapy	No vs. Yes	1.807 (1.170–2.792)	0.008	0.007
Histological grade	WD/MD vs. PD	1.742 (1.126–2.696)	0.013	0.012
EGFR mutation	No vs. Yes	0.482 (0.303–0.768)	0.002	0.002
ALK mutation	No vs. Yes	0.523 (0.191–1.429)	0.206	0.198
Pathological T stage	T_mi_/T1			0.006
	T2	1.593 (0.947–2.678)	0.079	
	T3	2.211 (1.143–4.277)	0.018	
	T4	3.360 (1.561–7.231)	0.002	
Pathological N stage	N0			<0.001
	N1	1.422 (0.759–2.665)	0.272	
	N2	2.872 (1.783–4.624)	<0.001	
Pathological substage	IA			<0.001
	IB	2.182 (0.861–5.530)	0.100	
	IIA	3.306 (1.081–10.113)	0.036	
	IIB	3.425 (1.500–7.828)	0.004	
	IIIA	6.210 (2.886–13.364)	<0.001	
	IIIB	8.464 (2.544–28.161)	<0.001	
Pathological stage	I			<0.001
	II	2.375 (1.288–4.378)	0.006	
	III	4.461 (2.557–7.783)	<0.001	
SUVmax	Low vs. High	2.647 (1.463–4.789)	0.001	<0.001
SUVmax (continuous)		1.048 (1.006–1.091)	0.023	-
SUVmean	Low vs. High	2.226 (1.442–3.437)	<0.001	<0.001
SUVmean (continuous)		1.229 (1.061–1.424)	0.006	-
MTV	Low vs. High	3.151 (2.043–4.860)	<0.001	<0.001
MTV (continuous)		1.013 (1.007–1.019)	<0.001	-
TLG	Low vs. High	3.012 (1.947–4.661)	<0.001	<0.001
TLG (continuous)		1.001 (1.001–1.002)	0.001	-

HR, hazard ratio; CI, confidence interval; SUVmax, maximum standardized uptake value; SUVmean, mean standardized uptake value; MTV, metabolic tumor volume; TLG, total lesion glycolysis; WD, well differentiated; MD, moderately differentiated; PD, poorly differentiated.

**Table 3 cancers-15-04380-t003:** Multivariate Cox regression analysis for overall survival including MTV and TLG in patients with operable lung adenocarcinoma.

Variable	Categories	MTV		TLG	
		HR (95% CI)	*p*-Value	HR (95% CI)	*p*-Value
Sex	Female vs. Male	1.286 (0.789–2.097)	0.313	1.395 (0.864–2.253)	0.173
Age, years	<58				
	58–67	1.688 (0.880–3.239)	0.116	1.572 (0.821–3.011)	0.172
	>67	3.267 (1.780–5.995)	<0.001	3.088 (1.679–5.677)	<0.001
Adjuvant therapy	No vs. Yes	0.730 (0.418–1.277)	0.270	0.744 (0.426–1.301)	0.299
Histological grade	WD/MD vs. PD	1.338 (0.853–2.097)	0.204	1.278 (0.809–2.019)	0.294
EGFR mutation	No vs. Yes	0.573 (0.348–0.945)	0.029	0.593 (0.361–0.976)	0.040
Pathological stage	I				
	II	2.074 (0.994–4.328)	0.052	2.113 (1.000–4.464)	0.050
	III	4.528 (2.216–9.254)	<0.001	4.661 (2.278–9.539)	<0.001
MTV	Low vs. High	2.043 (1.272–3.282)	0.003		
TLG	Low vs. High			1.761 (1.085–2.857)	0.022

HR, hazard ratio; CI, confidence interval; MTV, metabolic tumor volume; TLG, total lesion glycolysis; WD, well differentiated; MD, moderately differentiated; PD, poorly differentiated.

**Table 4 cancers-15-04380-t004:** HRs of overall survival in each subgroup based on pathological stage and MTV.

Pathological Stage	Low MTV			High MTV		
	HR (n)	95% CI	*p*-Value	HR (n)	95% CI	*p*-Value
I	- (178)	-	-	3.158 (17)	1.039–9.597	0.043
II	1.811 (67)	0.784–4.184	0.165	4.170 (55)	2.011–8.646	<0.001
III	3.680 (66)	1.830–7.401	<0.001	8.126 (49)	4.153–15.899	<0.001

HR, hazard ratio; CI, confidence interval; MTV, metabolic tumor volume.

## Data Availability

The data generated in this study are available upon request from the corresponding author.

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
