# Peer review of "Prognostic Significance of Volumetric Parameters Based on FDG PET/CT in Patients with Lung Adenocarcinoma Undergoing Curative Surgery"

_cancers, 2023, doi:10.3390/cancers15174380_

Round 1

Reviewer 1 Report

[Cancers] Manuscript ID: cancers-2583856

This is very interesting report showing that MTV or TLG in FDG-PET/CT of lung adenocarcinoma was independent prognostic factor after surgery. They have tried a new staging system using PET parameters and it works better than the conventional system.

I hope the authors can address several unclear points and limitation.

1.    Abstract, Line 38. 0.698 vs 0.673 : Were they statistically significant ? Please provide information.

2.    Subjects. They have excluded 385 subjects (44%) having SUVmax less than 2.5.

One of the important characteristics of adenocarcinoma is having low FDG uptake. And these 44% seems to be a population of good prognosis. When I considered this, exclusion of 44% of low FDG uptake subjects from the analysis may affect some kind of bias to their results. Please consider these factors a bit more as a limitation of the study.

3.    Results. Line 255. The new staging system had higher AUC values than the conventional staging system. Was it statistically significant?

Good.

Reviewer 2 Report

Thank you for the opportunity to review the article entitled:

Prognostic significance of volumetric parameters based on FDG PET/CT in patients with lung adenocarcinoma undergoing curative surgery

General comments:

The article is interesting and based on original data, therefore, deserves attention. The article demands language revision and making sure the appropriate manner of describing both utilities and radiopharmaceuticals (i.e. 18F-FDG, providing full chemical naming upfront). It is important the make sure that the scientific article is detailed, precise and leaving no room for general and vague characteristics.

Specific comments:

Abstract: the abstract seems accurate and describing the most relevant for this summary data. One remark: please, reconsider the conclusion. Although the results seem concluded, it would be much more useful for the Readers to know what does it mean when the measured parameters are “independent” (clinical conclusion, practical dimension of the study outcome).

Introduction: The introduction section is probably the most suitable for the appropriate background place in the article. The 18F-FDG PET-CT study has been recognized as useful imaging tool in lung cancer diagnosis in every stage of the disease, especially when dual-time-point (DTP) studies are considered (issue which has been studied for years). Please, provide the following information adding a paragraph and consider citing articles in which the DTP methodology in various diagnoses has been studied in detail to make sure that the Introduction is full and serves both as scientific and educational material inclusively for all Readers:

https://doi.org/10.3389/fonc.2021.559623

https://doi.org/10.3390/diagnostics10100836

https://doi.org/10.1016/j.crad.2010.10.008

https://doi.org/10.1038/s41598-020-59832-4

Furthermore, please provide recent epidemiological statistics, follow World Cancer Reports and make sure you characterized the worldwide tendencies in lung ca occurrence. It is important to enhance the value of obtained results in the group of tumours that are clinical issue. It definitely highlights how important is to study this particular group of patients.

Moreover, it would be extremely valuable to provide therapy options background to make sure that the Readers have full picture both of diagnostic methods that can be applied to lung ca patients and treatment options, accordingly with the international clinical standards.

Methods: The sample-size is worth attention. Large cohorts provide more valuable data. I appreciate the inclusion/exclusion criteria provided in flow chart. Please comment on the in subcohorts sample-sizes differences influence on the final results and conclusions. Please consider shortening the Figure 1 title and adding the legend or providing the commentary beneath the Figure instead of commentary within the title. Please, make sure you supported the acquisition protocol design with guidelines (i.e. EANM). Please, consider providing technical conditionings in table. Please, provide the information about the patients in terms of diagnostic pathway they underwent (have they underwent any other studies except by 18F-FDG PET/CT?). Please, provide the statistical software data.

Results: Please, place patients characteristics in Methods section and make sure not to overly repeat these data within the manuscript. That being said, 3.1. section should be included in Methods along with the study design. Please, make sure you provided a little bit of theoretical background to all of used tools to ensure the educational aspect of the scientific material. The article is interesting, using multiple tools that shall be highlighted and discussed in the Discussion section. Make sure that all parameters are relevant for the study. Please, filter the numeric results and show which are the most important for the study outcome. Please, make sure that Figure 2 is self-explanatory in full. Appropriate commentary is relevant for the Reader to understand the clinical impact of the obtained results.

Discussion: The study is full of data but difficult to follow. Please, make sure that you explained in the Discussion section the following issues:

-        Why the study is relevant to the field?

-        Why the Authors conducted the study? What were their hypotheses?

-        What tools have been applied to the study design in order to test hypotheses?

-        Which parameters have been the most relevant for the study?

-        Which findings have been concluded the most relevant?

-        Which are the study limitations and how to solve them?

-        What is the perspective of the study? Will it be continued in any way? What is missing in the report?
